# Epstein-Barr Virus BARF1 Is Expressed in Lung Cancer and Is Associated with Cancer Progression

**DOI:** 10.3390/cells13181578

**Published:** 2024-09-19

**Authors:** Julio C. Osorio, Alvaro Armijo, Francisco J. Carvajal, Alejandro H. Corvalán, Andrés Castillo, Ezequiel M. Fuentes-Pananá, Carolina Moreno-León, Carmen Romero, Francisco Aguayo

**Affiliations:** 1Laboratorio de Oncovirología, Departamento de Ciencias Biomédicas, Facultad de Medicina, Universidad de Tarapacá, Arica 1000000, Chile; jcosoriop@academicos.uta.cl (J.C.O.); alvaro.armijocr@gmail.com (A.A.); carolinajohanamoreno@gmail.com (C.M.-L.); 2Laboratory of Endocrinology and Reproductive Biology, Hospital Clínico Universidad de Chile, Santiago 8380456, Chile; 3Department of Hematology and Oncology, School of Medicine and Advanced Center for Chronic Diseases, Pontificia Universidad Católica de Chile, Santiago 8320000, Chile; francisco.carvajalc@outlook.cl (F.J.C.); acorvalan@uc.cl (A.H.C.); 4Department of Biology, Faculty of Natural and Exact Sciences, Universidad del Valle, Cali 760032, Colombia; andres.castillo.g@univalle.edu.co; 5Unidad de Investigación en Virología y Cáncer, Hospital Infantil de México Federico Gómez, Mexico City 06720, Mexico; ezequiel.fuentes@alumni.bcm.edu

**Keywords:** lung, cancer, EBV

## Abstract

Background: Epstein–Barr virus (EBV) is involved in the development of lymphomas, nasopharyngeal carcinomas (NPC), and a subgroup of gastric carcinomas (GC), and has also been detected in lung carcinomas, even though the role of the virus in this malignancy has not yet been established. BamH1-A Rightward Frame 1 (BARF1), a suggested exclusive epithelial EBV oncoprotein, is detected in both EBV-associated GCs (EBVaGC) and NPC. The expression and role of BARF1 in lung cancer is unknown. Methods: A total of 158 lung carcinomas including 80 adenocarcinomas (AdCs) and 78 squamous cell carcinomas (SQCs) from Chilean patients were analyzed for EBV presence via polymerase chain reaction (PCR), Immunohistochemistry (IHC), or chromogenic in situ hybridization (CISH). The expression of BARF1 was evaluated using Reverse Transcription Real-Time PCR (RT-qPCR). Additionally, A549 and BEAS-2B lung epithelial cells were transfected with a construct for ectopic BARF1 expression. Cell proliferation, migration, invasion, and epithelial–mesenchymal transition (EMT) were evaluated. Results: We found that EBV was present in 37 out of 158 (23%) lung carcinomas using PCR. Considering EBV-positive specimens using PCR, IHC for Epstein–Barr nuclear antigen 1 (EBNA1) detected EBV in 24 out of 30 (80%) cases, while EBERs were detected using CISH in 13 out of 16 (81%) cases. Overall, 13 out of 158 (8%) lung carcinomas were shown to be EBV-positive using PCR/IHC/CISH. BARF1 transcripts were detected in 6 out of 13 (46%) EBV-positive lung carcinomas using RT qPCR. Finally, lung cells ectopically expressing BARF1 showed increased migration, invasion, and EMT. Conclusions. EBV is frequently found in lung carcinomas from Chile with the expression of BARF1 in a significant subset of cases, suggesting that this viral protein may be involved in EBV-associated lung cancer progression.

## 1. Introduction

Lung cancer is a very heterogeneous malignancy currently classified into two types: small cell lung carcinomas (SCLCs) and non-small cell lung carcinoma (NSCLCs) [1]. Lung cancer is the most fatal (1.8 million deaths) and most frequently diagnosed (2.21 million cases) cancer worldwide [2]. Tobacco smoking (TS) is the most significant risk factor for lung cancer development. Additional risk factors include secondhand smoking, a family history of lung cancer, and exposure to chemicals such as radon, arsenic, and asbestos [3]. Interestingly, viral infections have been found in a subset of lung carcinomas, although an etiological role has not yet been established [4]. 

Epstein–Barr virus (EBV) is a herpesvirus that has been detected in a variable subset of lung carcinomas [5], with demographic variations. EBV is currently present in ~95% of the human population, establishing persistent infections for the lifetime of the host [6]. EBV preferentially infects both B cells and epithelial cells [7] and has been characterized as a class I carcinogen by the International Agency for Research on Cancer (IARC) [8]. In B cells, EBV establishes latency with viral reactivation when these cells differentiate into plasma cells. However, in normal epithelial cells, EBV seems to, by default, activate the lytic cycle, leading to viral maturation and release [9]. Interestingly, the expression of latency genes such as Epstein–Barr nuclear antigen 1 (EBNA1) and Epstein–Barr small RNAs (EBERs) is commonly observed in EBV-associated carcinomas including gastric cancer (EBVaGC) and nasopharyngeal carcinoma (NPC), where the virus establishes a type I/II latency [10,11]. Although the mechanisms of EBV-mediated carcinogenesis are not fully understood, multiple alterations are involved, including the expression of EBV and cellular oncogenes, viral and host epigenetic changes, nuclear factor- κB (NF-κB), Phosphatidyl 3-kinase (PI3K)/Akt, Janus kinase (JAK)-signal transducer/activator of transcription (STAT), and Mitogen-activated protein kinase (MAPK) pathway activation. Interestingly, evidence demonstrates that lytic genes are also expressed in EBV-associated cancers [12,13], including BamHI-A Rightward Frame 1 protein (BARF1), an early lytic protein frequently detected in EBV-positive carcinomas [14]. Indeed, BARF1 inhibits the activation of macrophages by binding to the granulocyte Macrophage Colony-Stimulating Factor (GM-CSF), thus contributing to viral immune evasion [15]. Importantly, BARF1 promotes cell proliferation and shows anti-apoptotic properties, suggesting a contribution to the oncogenic potential of EBV [16]. In this study, we addressed both the presence of EBV and the BARF1 expression in lung carcinomas from Chilean patients. Additionally, we characterized the role of BARF1 in lung cancer progression using in vitro approaches.

## 2. Materials and Methods

### 2.1. Clinical Specimens 

This is a retrospective study. One-hundred and fifty-eight formalin fixed paraffin embedded (FFPE) lung carcinomas from Chilean patients were collected for this study between 2012 and 2016. The samples were 80 adenocarcinomas (AdCs) and 78 squamous cell carcinomas (SQCs), obtained from the Pathological Anatomy Service of the National Thorax Institute, Santiago, Chile. Additionally, the mean age of patients was 66.7 years, ranging between 35 and 95 years old. The clinical data of the patients were obtained from the pathology reports and the patients’ clinical records of the same hospital. The Ethics Committee Board of the University of Chile approved this study (N° 027; 25 May 2022). 

### 2.2. Cell Culture and Stable Transfections

A549 (lung adenocarcinoma, CCL-185) cells and BEAS-2B (normal bronchial tissue, CRL-9609) cells were obtained from the American Type Culture Collection (ATCC) (Manassas, VA, USA) [17,18]. The A549 cell line was maintained in RPMI-1640 (Invitrogen, Carlsbad, CA, USA), whereas the BEAS-2B cell line was maintained in Dulbecco-modified Eagle medium (DMEM). Both cell lines were supplemented with 10% inactivated fetal bovine serum (FBS) (Hyclone), 0.1 μg/mL gentamicin (Invitrogen, Carlsbad, CA, USA), 1 U/mL penicillin, and 1 μg/mL streptomycin (Invitrogen, Carlsbad, CA, USA), and incubated at 37 °C in a 5% CO_2_ atmosphere incubator according to ATCC instructions. The cell lines were checked for mycoplasma infection using standardized methods. Cells were seeded into 6-well plates at a density of 1.5 × 10^5^ cells per well and, on the following day, were transfected with 1.5 μg MSCV (Addgene, Plasmid #41033, depositing name: Wade Harpe) or MSCVBARF1 (Addgene, Plasmid #37922, depositing lab: Karl Munger) plasmids using FuGENE^®^6 transfection reagent (Promega, Madison, WI, USA) following the manufacturer’s protocol. Transfected A549 and BEAS-2B cells were maintained in culture medium without antibiotics for 12–18 h and selected with 0.3 µg/mL puromycin for two weeks (Gibco, Carlsbad, CA, USA), as previously described in Muñoz JP et al. (2012) [18] and Peña N et al. (2015) [17]. 

### 2.3. DNA Extraction and EBV Detection 

FFPE tissues were incubated with digestion buffer as previously described [18]: (10 mM Tris-HCl pH 7.4; 0.5 mg/mL proteinase K, and 0.4% Tween 20) at 56 °C for 8 h with continuous stirring. After digestion, the samples were incubated at 95 °C for 10 min followed by centrifugation for 2 min at 16,000× *g*. The samples were maintained at −4 °C. The detection of viral DNA was performed using conventional end-point polymerase chain reaction (PCR) as previously described [19]. Firstly, the amplification of a fragment of the human β-globin gene was used to determine the quality of the obtained DNA. The sequence of primers was as follows: PCO3: 5′-ACACAACTGTGTTCACTAGC-3′ and PCO4: 5′-CAACTTCATCCACGTTCACC-3′. The following amplification program was used: initial denaturation at 95 °C for 5 min; 45 cycles with a cycling profile at 95 °C for 30 s, 52 °C for 30 s, and 72 °C for 30 s; and final extension at 72 °C for 5 min. To determine the presence of EBV genomes, we used consensus primers to amplify a fragment of the BNRF1 gene. The primer sequence was as follows: F: 5′-CCTGGTCATCCTTTGCCA-3′ and R: 5′-TGCTTCGTTATAGCCGTAGT-3′. The following PCR conditions were used as previously described [20]: denaturation at 95 °C for 5 min; 40 cycles that included 30 s at 95 °C, 30 s at 60 °C, and 30 s at 72 °C; and final extension at 72 °C for 5 min. Amplification products were characterized by 3% agarose gel electrophoresis, stained with the reagent SafeView PlusTM (abm, Vancouver, BC, Canada) and exposed to UV transillumination for visualization. 

### 2.4. Reverse Transcription Real-Time PCR 

RT-qPCR was carried out according to previously established protocols [17,18,21]. Total RNA obtained from tissues was purified using the High Pure FFPET RNA Isolation Kit (Roche Molecular Systems, Inc., Pleasanton, CA, USA) according to the protocol established by the manufacturer. RNA samples were mixed with a nonspecific RNA (free of EBV transcripts) to complete 1 µg. For cDNA synthesis, a reaction mixture containing 1 µg RNA, 1 U RNase inhibitor (Promega Corp., Madison, WI, USA), 0.2 µg random primers (Promega, Madison, WI, USA), 2 mM dNTP mix (Promega, Madison, WI, USA), and 10 U Moloney Murine Leukemia Virus reverse transcriptase (M-MLV RT) (Promega Corp., Madison, WI, USA), in a final reaction mixture of 20 µL, was incubated at 37 °C for 1 h and maintained at −20 °C until use. MMLV negative controls were included in each processing batch. For cDNA amplification, the reaction mixture contained 12.5 µL of 2X GoTaq^®^ G2 Green Master Mix (Promega, Madison, WI, USA), 20 µM forward and reverse primers, 10.5 µL RNase-free water (Promega Corp., Madison, WI, USA), and 1 µL of target cDNA. Endogenous β-actin mRNA levels were used for the normalization of RNA expression. For β-actin, the amplification conditions were 94 °C for 5 min, followed by 40 cycles at 95 °C for 15 s, and annealing–extension at 60 °C for 15 s. A dissociation curve was generated after each amplification protocol. The primer sequence was as follows: forward: 5′-CCACACTGTGCCCATCTACG-3′ and reverse: 5′-AGGATCTTCATGAGGTAGTCAGTCAG-3′. To identify the presence of EBV transcripts, we used the following primers: BARF1-F: 5′-CTTTCTTGGGTGAGCAGCGAGTC-3′ and BARF1-R: 5′-CAAATAAGCACCTGCTCCTC-3′. The following real time PCR conditions were used: denaturation at 95 °C for 10 min; 40 cycles that included 15 s at 95 °C, 15 s at 53 °C, and 15 s at 72 °C; the melting program was 30 s at 95 °C, 30 s at 65 °C, and 30 s at 95 °C. 

### 2.5. Immunohistochemistry for EBV EBNA1

Immunohistochemistry (IHC) for EBNA1 protein in 4% FFPE tissues was performed according to previously established protocol [19]. Three µm-thick histological sections were prepared and left to dry overnight at 55 °C, then were deparaffinized and hydrated with distilled water. For antigenic recovery, incubation with Tris-Borate-EDTA (TBE) buffer pH 8.0 at 95 °C for 8 min was used. The primary EBNA1 antibody was diluted 1:1500 and incubated at 37 °C for 16 min. The Novolink Polymer DS detection system (Leica Biosystems, Richmond, IL, USA (RE7140-K)) was used according to the supplier’s instructions, using diaminobenzidine (DAB) as chromogen. The results for EBNA1 IHC were checked by an experienced pathologist and were informed as negative or positive according to the guidelines established by the College of American Pathologists (positive specimen ≥ 70% nuclear staining).

### 2.6. Chromogenic In Situ Hybridization for EBV 

Epstein–Barr encoded early RNAs (EBERs) were detected using chromogenic in situ hybridization (CISH) with the ZytoFast EBV Probe (Digoxigenin-labeled) reagent and the ZytoFast PLUS CISH Implementation Kit (HRP-DAB) (ZytoVision, Bremerhaven, Germany) following the manufacturer’s instructions and according to previously established protocol [22]. Additional modifications were realized: (1) denaturation was carried out using a pressure cooker and 10 µL probe solution was applied to each clinical sample. For the verification of cellular mRNA integrity, the ZytoFast 28S rRNA (+) control probe was used. Raji cells (cell line obtained from a patient with Burkitt’s lymphoma) were utilized as positive controls in each processing batch. A result was considered positive when a blue color localized in the cancer epithelial cells was observed. The results were interpreted by an experienced pathologist.

### 2.7. Cell Viability

To perform cell proliferation, assays were performed according to previously established protocol [18], and CCK-8 reagent was used. A549 cells were grown in RPMI-1640 medium while BEAS-2B cells were grown in DMEM in 60 mm dishes for 24 h. The cells were trypsinized using 1X EDTA–trypsin mixture (Invitrogen, Carlsbad, CA, USA) and then counted using Trypan blue staining in a Neubauer chamber. Three 96-well plates with 3000 cells were cultured in triplicate in 100 μL of culture medium (RPMI for A549 cells and DMEM for BEAS-2B cells). After 24, 48, and 72 h of culture, CCK8 reagent was added in a 1:10 dilution in culture medium. After 4 h at 37 °C, the colorimetric measurement was carried out at 450 nm in a spectrophotometer.

### 2.8. Cell Migration

Cell migration was carried out using Boyden chamber assay (transwells) as previously published [23]. The bottom side of transwell upper chambers (Corning, New York, NY, USA) was briefly coated with 3 µg/mL fibronectin (Thermo Fisher Scientific, Inc., Waltham, MA, USA) and maintained at 4 °C overnight. Next, either 1.5 × 10^5^ A549 cells or 4 × 10^5^ BEAS-2B cells were seeded inside the transwell in a volume of 200 µL of serum-free RPMI-1640 media, and 500 µL of RPMI-1640 containing 10% FBS was added to each well. Cells were then incubated for 4 h at 37 °C and 5% CO_2_. Migrated cells were then fixed and stained in 0.5% crystal violet/20% methanol solution for 1 h at room temperature. Unmigrated cells were scraped from the upper chambers using cotton swabs, whereas migrated cells were counted in seven high-power fields (400×). Three independent experiments were carried out.

### 2.9. Cell Invasion

Cell invasion was carried out using upper chambers of transwells with 8-µm membrane pores as previously published [24]. Additionally, the membrane was pre-coated with Matrigel matrix gel (60 µL) (BD Biosciences, New Jersey, USA) through the application of culture medium (RPMI for A549 cells or DMEM for BEAS-2B cells) at 37 °C at least 2 h prior to the seeding of the tested cells. Subsequently, either 5 × 10^4^ A549 cells or 4.5 × 10^4^ BEAS-2B cells were seeded in the upper chamber of the transwell in 200 µL of SFB-free culture medium. In the lower chamber of the transwell, 500 μL of culture medium supplemented with 10% SFB were added as a chemoattractant. The Matrigel invasion chamber was deposited in a humidified tissue culture incubator for 24 h. Then, the upper chambers were removed from the lower chambers and immediately wiped using cotton swabs. The invaded and migrated cells were fixed using 100% methanol for 15 min at room temperature, then visualized and quantified using toluidine blue. Ten fields of each chamber were considered for analysis and photographed using an optical microscope (×40 magnification, Olympus Corp., Tokyo, Japan).

### 2.10. Epithelial Mesenchymal Transition Using Western Blotting

Epithelial–mesenchymal transition using Western blotting was performed according to previously established protocol [25]. Total protein from cells was extracted using a lysis buffer [20 mM Tris (pH 8.0), 1% SDS] that contained a protease inhibitor cocktail (Roche Diagnostics, Basel, Switzerland). The cell extracts were incubated at 4 °C for 1 h, sonicated at 20 kHz for 20 s on ice and immediately centrifuged at 12,000× *g* for 10 min at 4 °C. Consecutively, the proteins quantification was carried out using the Pierce Bicinchoninic Acid Protein Assay kit (Pierce; Thermo Fisher Scientific, Inc.). Thirty µg of protein extract was loaded per well and the proteins were separated via 12% sodium dodecyl sulfate (SDS) polyacrylamide gel electrophoresis (PAGE). Immediately, the proteins were transferred to Hybond-P ECL membranes (Amersham; GE Healthcare, Chicago, USA) using a buffer containing 20 mM Tris and 150 mM glycine (pH 8.3) in 20% methanol with a semi-dry transfer system (Bio-Rad Laboratories, Inc., Hercules, USA). Membranes were then incubated at room temperature for 2 h with the blocking reagent containing 5% bovine serum albumin (AppliChem GmbH, Darmstadt, Germany) in 0.5% Tris buffered saline/0.1% Tween-20 (TBST, pH 7.6) followed by an overnight incubation at room temperature with the following primary antibodies: β-actin (cat. no. ab6276; Abcam, Cambridge, UK), E-cadherin (cat. no. sc-21791; Santa Cruz, Dallas, USA), and vimentin (cat. no. sc-6260; Santa Cruz, Dallas, USA). These were diluted 1:1000 in TBST. The membranes were washed three times in TBST and incubated at room temperature for 1 h with peroxidase-labeled secondary IgG antibodies (Jackson Inmunoresearch code number: 115-035-003 (Anti-mouse) and Jackson Inmunoresearch code number: 111-035-03 (anti-rabbit)) that were diluted 1:5000 in TBST. After washing the membranes three times using TBST, immune complexes were identified using an ECL system (Amersham; GE Healthcare, Chicago, USA) according to the protocol established by the manufacturer. ImageJ software version 1.52a (National Institutes of Health) was used for semi-quantitative analysis.

### 2.11. Statistical Analysis

The statistical analysis was carried out using the GraphPad software. Additionally, Chi-square and Fisher’s exact tests were used to analyze statistical significance. A *p* ≤ 0.05 was considered statistically significant. 

## 3. Results

### 3.1. EBV Presence in Lung Carcinomas Using Conventional PCR 

In this study, 80 lung AdCs and 78 lung SQCs from Chilean patients were analyzed. SQCs were significantly more frequent among smokers than AdCs (*p* = 0.046). Additionally, a significant difference was found in differentiation grade among SQCs and AdCs (*p* = 0.047). No differences in age range were found (*p* = 0.498). The clinicopathological features of specimens used in this study are shown in Table 1.

All the specimens were positive for a fragment of the beta-globin gene, which is used as an inner control of amplifiable DNA presence. The presence of EBV was evaluated in these specimens using conventional PCR. A representative amplification of both the beta-globin and the EBV BNRF1 fragment is shown in Figure 1. 

EBV was detected in 23% (37/158) of lung carcinomas. No significant differences were found between EBV presence and age range (*p* = 0.999). Interestingly, EBV presence was more frequent in non-smoker patients when compared with smokers (33% vs 10%), although this difference was not statistically significant (*p* = 0.052). Regarding the differentiation grade of the tissue, there was a tendency to find EBV-positive samples in poorly differentiated tumors, though this difference was not statistically significant (*p* = 0.060). When the samples were stratified by histological type (80 AdCs and 78 SQCs), no statistical differences were found in EBV presence (*p* = 0.852) (Table 2).

### 3.2. EBV Presence in Lung Carcinomas Using EBNA1 Immunohistochemistry

Before evaluating the presence of EBNA, the histological characteristics of the tumor tissue were analyzed. The most frequent histological types in the EBV-positive lung carcinomas were the following: (1) a lepidic pattern that showed the lining of the alveoli with neoplastic pneumocytes; (2) a moderately differentiated acinar pattern where a tubular structure was noted; (3) a papillary pattern, where intraluminal papillary projections were observed; and (4) a solid pattern, which did not present tubules, papillary structures, or a lepidic pattern (Figure 2). Then, we evaluated the presence of EBNA1 using IHC in PCR-positive lung carcinomas. A total of 6 samples out of 37 EBV-positive lung carcinomas were used up, and only 31 PCR-positive specimens were analyzed using IHC. We found that 30/31 (97%) of PCR-positive lung carcinomas were IHC positive. Considering the 30 IHC-positive specimens, 80% (24/30) showed an EBNA1 signal in the tumor cells while in 20% (6/30) all the EBNA1 signals were found in stromal cells rather than tumor cells. Regarding the pattern type, EBV was present in 28% of the solid type, while in the papillary and trabecular types it was present in 13% of each. When samples were stratified by histological type (15 AdCs and 15 SQCs), no statistical differences were found and the EBNA1 presence was approximately 41% in each histological type (Table 3, Figure 3 and Appendix A). 

### 3.3. EBV Presence in Lung Carcinomas Using EBERs CISH

We evaluated the presence of EBERs using CISH in PCR-positive lung carcinomas. Previously, we had only been able to process 30 samples positive for EBNA using IHC, and an additional 14 samples ran out of material in the paraffin block. We found that 13/16 (81%) of the PCR-positive lung carcinomas tested were EBERs positive. Furthermore, 9 of 16 (56%) samples also expressed EBNA-1 in nuclei (Table 3, Figure 4 and Appendix A). Furthermore, we analyzed clinical characteristics of lung cancer tumors with the EBV presence using CISH. Overall, EBV was detected in 8% (13/158) of lung carcinomas using PCR/IHC/CISH. A statistically significant association between EBV presence and age range (*p* = 0.02358) was found. A non-statistically significant association between non-smoker patients and smokers (*p* = 0.2913) was found. No significant statistical differences between poorly, moderate, and well differentiated tumors were found (*p* = 0.6708). When the samples were stratified by histological type, no statistical differences were found in EBV presence (*p* = 0.2987) (Table 4).

### 3.4. BARF1 Detection in Lung Carcinomas Using RT-qPCR

BARF1 transcript detection was carried out using RT-qPCR, finding BARF1 transcripts in 19% (7/37) of EBV-positive lung carcinomas. Overall, 13 out of 152 (9%) lung carcinomas were shown to be EBV-positive using PCR/IHC/CISH. A total of 7 of these 13 samples also exhibited BARF1 expression. Additionally, when the samples were stratified by histological type (15 SQCs and 15 AdCs), BARF1 expression was not statistically different (21% in SQCs and 17% in AdCs, *p* = 0.758). Neither smoking habit (*p* = 0.585) nor differentiation status (*p* = 0.464) were statistically different among BARF1 positive and BARF1 negative cases. Regarding differentiation status, we found that BARF1 was detected in tissues with poor or moderate differentiation in both histological types of lung cancer (SQCs and AdCs) (Table 5). 

### 3.5. Proliferation of Lung Cells Ectopically Expressing BARF1 

We evaluated changes in proliferation of lung cancer and non-cancerous cell lines that stably express BARF1. Using RT-PCR, we found that only cells transfected with the MSCVBARF1 construct express BARF1 transcripts (Appendix A). Figure 5A shows no differences between A549 cells that express BARF1 and those transfected with the empty vector. However, the non-cancerous BEAS-2B cell line transfected with the BARF1 vector showed increased proliferation with respect to the control at 72 h. This allows us to conclude that the BARF1 expression correlates with an increased proliferation rate of non-tumor lung cells in vitro (Figure 5B).

### 3.6. Migration of Lung-Derived Cell Lines Ectopically Expressing BARF1 

Migration assays of the lung cancerous and non-cancerous cell lines were performed using transwells. Ectopic BARF1 expression significantly increased the migratory capacity of both A549 and BEAS-2B cells, the former increasing migration by 2-fold (Figure 6A,B), and the latter by 1.5-fold (Figure 6C,D).

### 3.7. Invasion of Lung-Derived Cell Lines Ectopically Expressing BARF1

The invasion capacity of the cell lines was determined using Matrigel/transwell. BARF1 expression was associated with a significant increase in invasion capacity of A549 cells when compared to the control (Figure 7A,B). Surprisingly, the expression of BARF1 significantly reduced the invading capacity of BEAS-2B cells by almost 2-fold (Figure 7C,D).

### 3.8. Epithelial Mesenchymal Transition in Lung-Derived Cell Lines Ectopically Expressing BARF1 

We determined changes in the levels of E-cadherin and vimentin, proteins involved in the epithelial–mesenchymal transition (EMT), using Western blotting. We observed a significant decrease in E-cadherin levels and an increase in vimentin levels in A549 cells that express BARF1 when compared with cells transfected with the empty vector (Figure 8A,B). However, E-cadherin levels were not detectable at basal levels and vimentin levels were not altered in the presence of BARF1 in BEAS-2B cells (Figure 8C,D).

In summary, we provide evidence of EBV positivity in a subfraction of lung carcinomas, in which 37 out of 158 (23%) samples were positive using PCR. Of these PCR-positive samples, 13 were also positive for the expression of EBNA1 and the EBERs. Those EBV-positive samples were enriched in patients with non-smoking habits and less differentiated tumor types. Furthermore, we observe that 7/37 of the PCR-positive lung tumors express BARF1 and provide in vitro evidence that ectopic BARF1 expression promotes an enhanced migratory and invasive capacity, correlating with the enrichment of an EMT phenotype.

## 4. Discussion

Tobacco smoking is the leading risk factor for lung cancer development [17]. However, a variable subset of lung carcinomas arises in non-smoker patients. Suggested factors potentially involved in lung cancer are viral infections such as high-risk human papillomaviruses (HR-HPVs) [26], Merkel cell polyomavirus (MCPyV) [27], Jaagsiekte Sheep Retrovirus (JSRV) [28], John Cunningham Virus (JCV) [29], and EBV [30]. Additionally, the virome data of normal lung tissues revealed the presence of 11 distinct viral DNA sequences. Indeed, human herpesvirus 6B (HHV-6B) and parvovirus B19V exhibited the highest detection rates (87% and 84%, respectively), followed by HHV-7 (61%), EBV (58%), and torque teno virus (TTV) (52%). Interestingly, both EBV and human cytomegalovirus (HCMV) demonstrated significantly higher lung prevalence (23%) than other tissues [31]. Additionally, human herpesviruses (HHVs), human adenovirus, human rhinoviruses, and influenza virus A were found in breath samples of children from China with serious respiratory illnesses using advanced genetic analysis [32]. Alternatively, high levels of RNA reads have been found for herpes simplex virus 1 (HSV-1), HCMV, EBV, and anellovirus in patients with non-respiratory symptoms using RNA sequencing-based metagenomics, raising questions about their potential role in the observed clinical presentation [33].

The presence of EBV in lung cancer is highly variable according to the tumor histological type and the geographical site. Indeed, EBV is frequently detected in pulmonary lymphoepithelioma-like carcinoma (LELC), principally from East and Southeast Asian countries. Although it is possible to find EBV in clinical specimens of lung cancer, the causal relationship between lung cancer and EBV remains unclear [34]. In this study, EBV was detected in 23% of lung carcinomas from Chilean patients with non-statistically significant differences between AdCs and SQCs. This study is the first known survey of EBV in lung cancers in South America. Additionally, EBV has been detected in 14.6% of lung carcinomas from the U.S. using qPCR and microarrays [35], though two additional reports from the same country detected EBV in only 0.3% and 2.7% of the cases, respectively [5,36]. In Asia, the prevalence of EBV in lung cancer is highly variable with frequencies ranging from 5.4% to 58.8% [34]. A Chinese study found 25 out of 48 lung carcinomas positive for EBV DNA and 75% of them were SQCs. Additionally, p53, Bcl-2, and c-myc gene expression was closely related to the presence of EBV DNA, with a significantly high expression in EBV-positive patients (*p* < 0.05) [37]. In Europe (Spain), researchers described 12 lung carcinomas (4 SQCs and 8 ADCs) with EBV sequences in the tumor tissue, detected using PCR and/or EBER ISH [38]. An Italian study evaluated the colonization of the airways using EBV in 70 lung carcinomas and 40 controls, analyzing the exhaled breath condensate using qPCR. This study found that 18/70 (26%) samples were positive for EBV [39]. Interestingly, another study from Italy detected EBV miRNAs in 7 out of 48 (15%) lung carcinomas using qPCR, even though only 1 of these 7 cases had detectable EBV DNA in the tumor tissue [35]. Importantly, a recent meta-analysis carried out considering 886 lung cancer patients found that the overall prevalence of EBV infection was 44.36% (95% CI: 4.08–16.9). Additionally, EBV presence increased the risk of lung cancer by a factor of four [30]. Thus, considering previous reports, the EBV frequency found in lung carcinomas from Chile is approximately half the average reported in different studies worldwide.

In our study, EBV showed increased frequency in poorly differentiated lung carcinomas (30%) when compared to highly differentiated cases (0%), though this difference was only close to significant (Table 1, *p* = 0.06), perhaps because of the low number of positive samples. It was suggested that poorly differentiated epithelial tumors mimic LELC [40], an entity strongly associated with EBV presence. Additionally, we observed that EBV was more frequent in non-smoker patients, even though this tendency was also close to significant (Table 1, *p* = 0.052). The high frequency of EBV in non-smoker patients could suggest the possibility that this virus works as an independent carcinogen in the lungs. Unfortunately, information about additional exposure risk factors in these patients was not available in the clinical records. Conversely, studies have established a strong link between smoking and EBV infection [41]. Indeed, smoking increases the risk of EBV reactivation from latency, which can contribute to various diseases such as NPC and multiple sclerosis [42]. This association is often attributed to the immunosuppressive effects of smoking, which could contribute to increased EBV reactivation [43]. 

Considering that PCR is a more sensitive method than IHC, we evaluated the presence of EBNA1 protein using IHC in PCR positive lung carcinomas. EBNA1 was detected in 97% of the EBV-positive specimens, in the epithelial cells. EBNA1 is an EBV protein expressed in all the latency programs (I-III) and is required for stable persistence of EBV [5,44,45]. As a DNA-binding protein, EBNA1 is involved in the episome segregation during mitosis and EBV genome tethering to the host genome through binding to specific sequences at OriP [46]. Thus, the identification of EBNA-1 expression in most of the EBV-positive cases suggest the possibility of a latent viral infection, even though additional analysis is required to establish the type of latency. Since it is important to determine whether lytic genes are also expressed during latency, we analyzed the expression of BARF1, an early lytic protein. As a result, BARF1 was detected in 19% of EBV-positive lung carcinomas. BARF1 specifically plays a crucial role in the development and progression of epithelial cancers [35]. A previous study found EBV in four lung carcinomas when ~1000 lung carcinomas and non-tumor adjacent tissues were analyzed using RNA sequencing. The specimen with the highest viral load showed an enrichment of transcripts from the BamHI-A region of the EBV genome that encodes *BARF1*. Another study focused on EBV strains derived from patients with pulmonary lymphoepithelioma-like carcinoma (pLELC) reported high-risk variations within the BWRF1, BILF2, BALF5, and BARF1 genes. These genes may be related to the malignant transformation ability of pulmonary epithelium of EBV [47,48]. Despite high conservation in the BARF1 gene sequence, studies have reported variation patterns. A study of northern Chinese individuals revealed 13 amino acid mutations, with V29A, V46A, D79G, V113I, and D138Y being the most prevalent. Interestingly, a statistically significant association was observed between the V29A mutation and NPC, compared to both EBVaGC and healthy control groups [49]. Of note, BARF1 expression is frequently detected in EBVaGC, though the role of this viral protein in gastric carcinogenesis is unclear [50]. 

The data in lung cell cultures showed that migration, invasion, and EMT were increased when A549 lung cancer cells ectopically expressed BARF1. An enhanced proliferation was also observed in the non-cancerous BEAS-2B lung cells. Interestingly, migration was also increased in BEAS-2B cells that express BARF1, even though the invasion ability was decreased and EMT was not altered. It is plausible that a differential genetic background in these tumor and non-tumor cell lines can explain these results. Interestingly, TP53 gene is expressed in both BEAS-2B cells [51] and A549 cells [52]. Indeed, it has been described that p53 can be involved in cancer metastasis regulation. In fact, p53 can prevent EMT, which may contribute to the induction of an EMT-like phenotype in p53-mutated tumors [53,54]. Additionally, p53 can promote MDM2-mediated degradation of Slug to increase E-cadherin expression [55]. Another plausible explanation is that BARF1 increases its effect when lung cells already have previous DNA damage or phenotypic alterations, suggesting that BARF1 may promote cancer cell metastasis [56]. It also has a high oncogenic potential associated with the different functions that have been characterized, such as cellular immortalization, increased cell proliferation, and immune evasion [57]. Regarding cellular immortalization, BARF1 can prevent cells from entering programmed cell death (apoptosis) by increasing the expression of anti-apoptotic proteins and decreasing the expression of pro-apoptotic proteins [58]. BARF1 can also increase cell proliferation by promoting the progression of the cell cycle from the G1 to S phase. This is achieved by increasing the expression of cyclin D1, a protein essential for cell cycle progression [59]. BARF1 is involved in immune system evasion by downregulating the expression of MHC class I molecules. BARF1 can increase cyclin D1 levels both at the transcript and protein levels [60]. In addition, experiments with EBV (-) cell lines have shown that the presence of BARF1 is associated with a lower expression of the tumor suppressor p21WAF1, an important cell cycle inhibitor [16]. Finally, we suggest a model in which BARF1 is expressed in EBV lung cancer cells and is involved in promoting lung cancer progression (Figure 9).

## 5. Conclusions

We have confirmed that EBV is present in 8% of lung carcinomas from Chilean patients through multiple techniques (PCR/IHC/CISH). Additionally, BARF1 was expressed in 7/13 of EBV-positive lung carcinomas. Importantly, BARF1 expression correlated with increased lung cancer cell migration, invasion, and EMT. This study suggests that EBV can be involved in a subset of lung carcinomas, with BARF1 working as a potential oncoprotein. More studies are warranted to address the roles of EBV and BARF1 in lung cancer.

## Figures and Tables

**Figure 1 cells-13-01578-f001:**
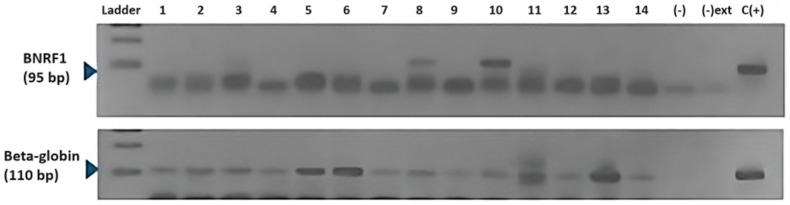
The 3% agarose gel electrophoresis of EBV BNRF1 (95 bp) and beta-globin (110 bp) gene fragments amplified using PCR. Numbers 1–14: clinical specimens; (-): PCR negative control; (-) ext: DNA extraction negative control; (+): positive control (Raji cells).

**Figure 2 cells-13-01578-f002:**
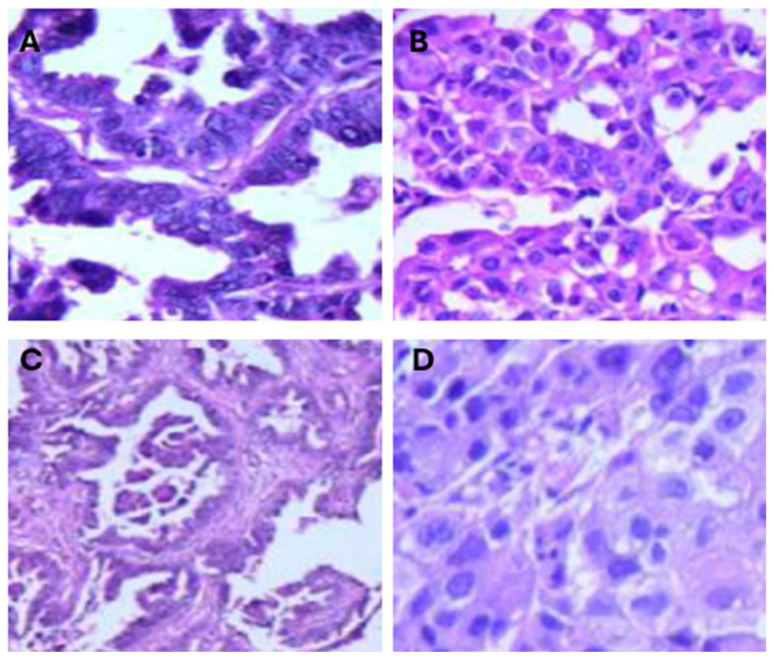
Histological types in lung carcinomas from Chile. (**A**) Lepidic pattern that shows the lining of the alveoli by neoplastic pneumocytes. (**B**) Moderately differentiated acinar pattern where a tubular structure is noted. (**C**) Papillary pattern, where intraluminal papillary projections are observed. (**D**) Solid pattern, which does not present tubules, papillary structures, or lepidic pattern.

**Figure 3 cells-13-01578-f003:**
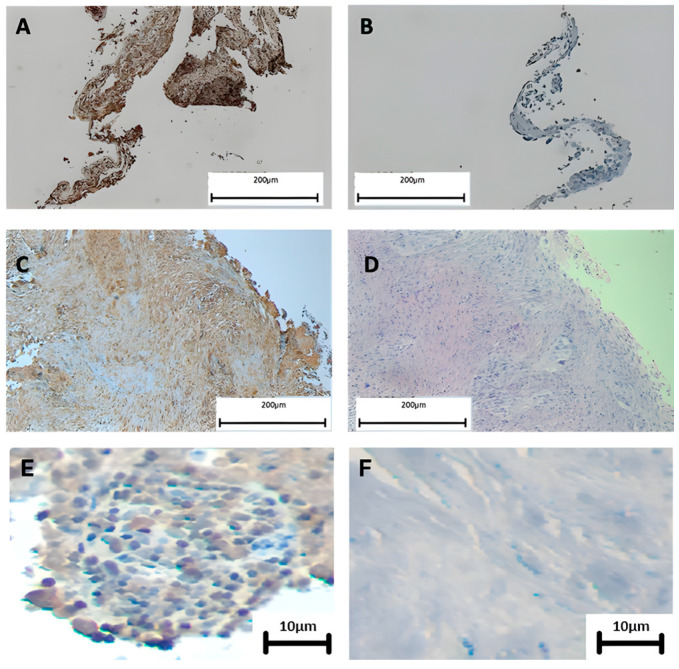
IHC for EBNA1 in lung carcinomas from Chile. (**A**) Undifferentiated nasopharyngeal carcinoma was used as the positive control; (**B**) the same tissue was used as the negative control without primary antibody; (**C**) lung carcinoma showing a positive signal for EBNA1 (brown color); (**D**) hematoxylin/eosin staining; (**E**) lung carcinoma showing a positive nuclear signal for EBNA1 (brown color); (**F**) lung carcinoma showing a negative nuclear signal for EBNA1.

**Figure 4 cells-13-01578-f004:**
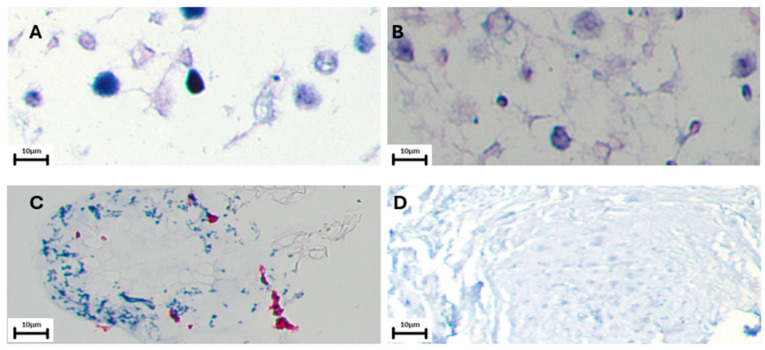
CISH for EBERs in lung carcinomas from Chile. (**A**) Raji cells were used as external positive control; (**B**) 28S rRNA positive control; (**C**) negative control with a random antibody. (**D**) Lung carcinoma showing a positive signal for EBERs (blue color). Black bar = 10 µm.

**Figure 5 cells-13-01578-f005:**
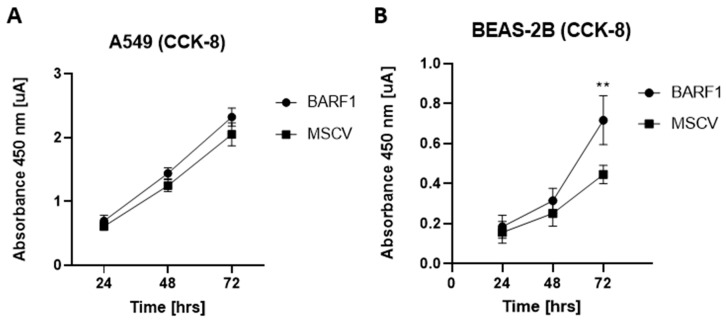
Proliferation assay in lung cells expressing BARF1. (**A**) Viability of A549 cells at 24, 48, and 72 h. (**B**) Viability of BEAS-2B cells at 24, 48, and 72 h. **: *p* ≤ 0.01. N = 3. Error bars correspond to SEM.

**Figure 6 cells-13-01578-f006:**
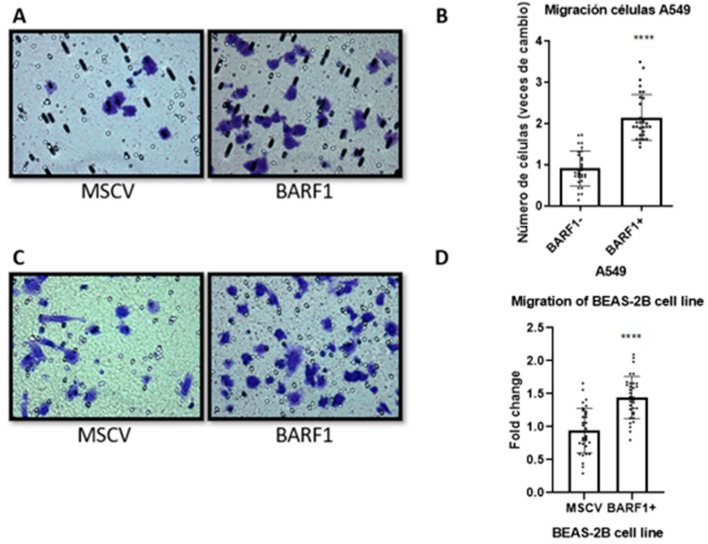
BARF1 expression is associated with the increased migration of lung-derived cell lines. (**A**) A549 cell migration assay photos were taken with a 400X magnification after 4 h of migration at 37 °C in an atmosphere of 5% CO_2_. (**B**) The result of the migration assay for A549 cells is presented as the ratio of change between migrating cells from the BARF1 condition over the average of migrating cells from the MSCV control condition (N = 3). (**C**) BEAS-2B cell migration assay photos were taken with a 40X magnification after 4 h of migration at 37 °C in an atmosphere of 5% CO_2_ for the BARF1 and MSCV conditions. (**D**) The result of the migration assay for BEAS-2B cells is presented as the ratio of change between migrating cells from the BARF1 condition over the average of migrating cells from the control condition (N = 3). ****: *p* ≤ 0.0001.

**Figure 7 cells-13-01578-f007:**
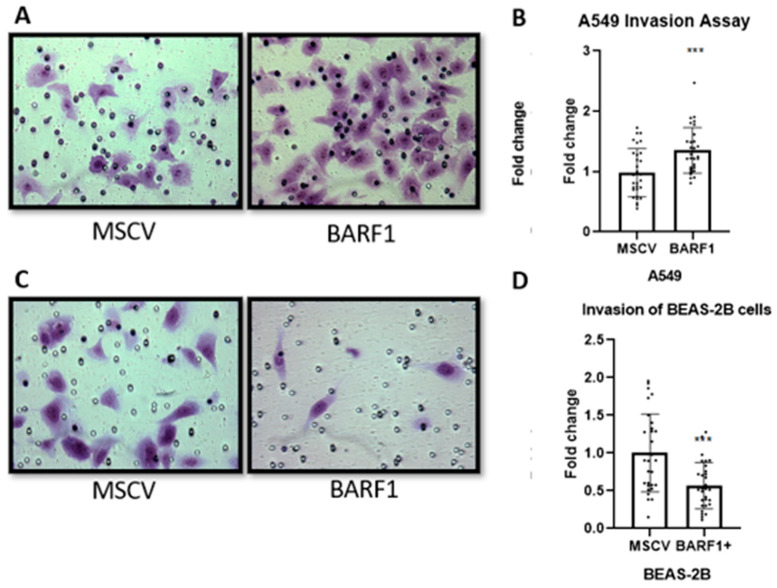
BARF1 expression is associated with increased invasion in A549 lung cancer cells. (**A**) A549 invasion assay using transwells carried out for 24 h in an atmosphere of 5% CO_2_. (**B**) The graph corresponds to the ratio of change between invading cells of the BARF1 condition over the average of the invading cells of the control condition (N = 3). (**C**) BEAS-2B invasion assay using transwells carried out for 24 h in an atmosphere of 5% CO_2_. (**D**) The graph corresponds to the ratio of change between invading cells of the BARF1 condition over the average of the invading cells of the control condition (N = 3). ***: *p* ≤ 0.001.

**Figure 8 cells-13-01578-f008:**
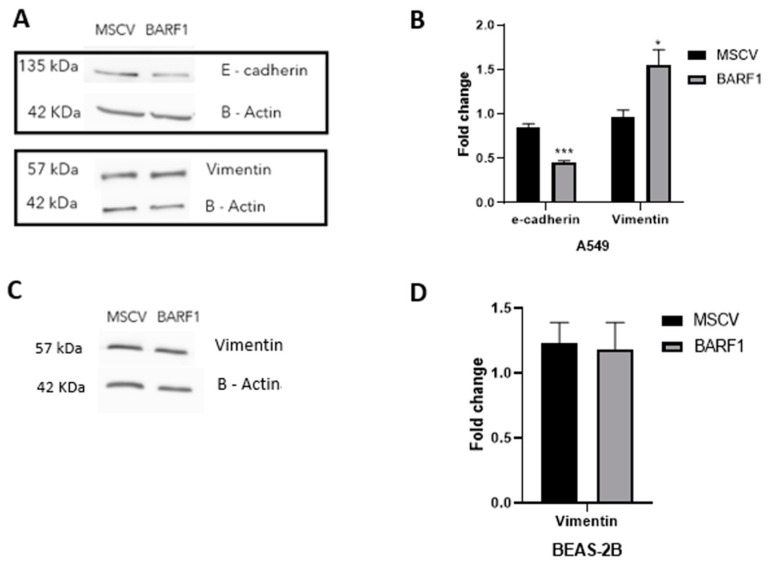
EMT biomarkers in lung cancer cells transfected with a BARF1 construct. (**A**,**B**) A549 tumor cells ectopically expressing BARF1 decrease E-cadherin protein levels detected using Western blot. (**C**,**D**) Non-tumor BEAS-2B cells do not change vimentin levels. *: *p* ≤ 0.05; ***: *p* ≤ 0.001.

**Figure 9 cells-13-01578-f009:**
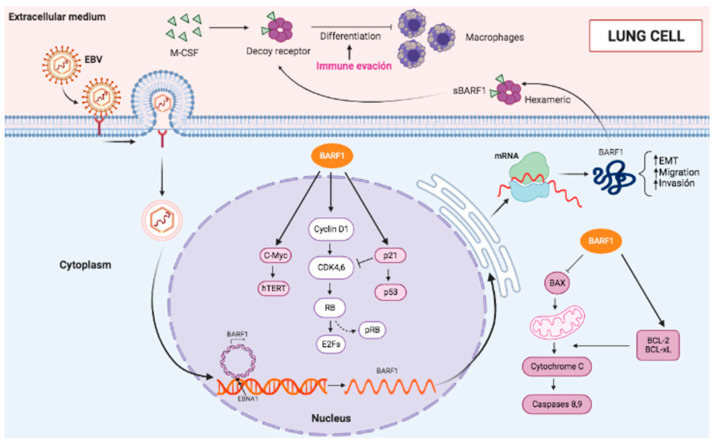
EBV BARF1 promotes increased cell proliferation, migration, invasion, and EMT in lung cancer cells. BARF1 is expressed as an early lytic protein with functions in immune cell evasion, cell immortalization, and lung cancer progression.

**Table 1 cells-13-01578-t001:** Clinicopathological features of lung SQCs and AdCs.

Histological Type	SQC	AdC	*p*-Value
	N (%)	N (%)	
Total	78 (49)	80 (51)	
Age			*p* = 0.498 *^,&^
≤65 years	23 (30)	28 (35)	
>65 years	55 (70)	52 (65)	
Differentiation			*p* = 0.047 *^,#^
Poor	49 (63)	47 (59)	
Moderate	29 (37)	27 (34)	
Well	0 (0)	6 (7)	
Smoking habit			*p* = 0.046 *^,#^
Smoking	25 (32)	17 (21)	
Non-smoking	1 (1)	7 (9)	
Without report	52 (67)	56 (70)	

^&^ Fisher’s exact test; ^#^ Chi Square test; *: *p* ≤ 0.05.

**Table 2 cells-13-01578-t002:** EBV presence in lung carcinomas from Chilean patients using PCR.

Feature	EBV Presence		Total	*p*-Value
	(-) Cases (%)	(+) Cases (%)		
TOTAL	121 (77)	37 (23)	158	
Age				
≤65 years	35 (69)	11 (31)	46	0.999 ^&^
>65 years	86 (77)	26 (23)	112	
Smoking habit				
Smoking	37 (90)	4 (10)	41	0.052 ^#^
Non-smoking	6 (67)	3 (33)	9	
Unknown	78 (72)	30 (28)	108	
Differentiation				
Poor	67 (71)	28 (29)	95	0.060 ^#^
Moderate	48 (84)	9 (16)	57	
Well	6 (100)	0 (0)	6	
Histology type				
SQC	59 (76)	19 (24)	78	0.852 ^&^
ADC	62 (77)	18 (23)	80	

^&^ Fisher’s exact test; ^#^ Chi Square test; *p* ≤ 0.05.

**Table 3 cells-13-01578-t003:** EBNA1 immunohistochemistry and CISH in EBV-positive lung carcinomas using PCR.

			EBV (-)	EBV (+)	Exhausted IHC	Exhausted CISH	*p*-Value
			N (%)	N (%)	N (%)	(N%)	
EBNA1/IHC	(N = 37)		1 (3)	30 (81)	6 (16)		
	Location						
		EBNA1 nuclear	1 (100)	24 (80)			0.8065 ^&^
		Stroma	0 (0)	6 (20)			
	Pattern type						
		Solid	0 (0)	8 (28)			0.9209 ^#^
		Papillary	0 (0)	4 (13)			
		Trabecular	0 (0)	4 (13)			
		Lipidic	0 (0)	1 (3)			
		Acinar	0 (0)	1 (3)			
		No pattern	1 (100)	12 (40)			
	Histological type						
	SQC		1 (100)	15 (50)			0.5161 ^&^
	AdC		0 (0)	15 (50)			
EBER/CISH	(N = 37)		3 (8)	13 (35)		21 (57)	
	Location						
		EBNA1 nuclear + EBER	3 (100)	9 (69)			0.3929 ^&^
		Other	0	4(31)			
	Histological type						
		SQC	1 (33)	6 (46)			0.5050 ^&^
		AdC	2 (67)	7 (54)			

^#^ Chi Square test for Table F by C; ^&^ Fisher’s exact test with mid-P method.

**Table 4 cells-13-01578-t004:** EBV presence in lung carcinomas from Chilean patients using EBERs.

Feature	EBERs EBV Presence		Total	*p*-Value
	(-) Cases (%)	(+) Cases (%)		
TOTAL	145 (92)	13 (8)	158	
Age				
≤65 years	69 (48)	3 (23)	72	0.02358 ^&^
>65 years	76 (52)	10 (77)	86	
Smoking habit				
Smoking	40 (98)	1 (2)	41	0.2913 ^#^
Non-smoking	8 (89)	1 (11)	9	
Unknown	97 (90)	11 (10)	108	
Differentiation				
Poor	89 (94)	6 (6)	95	0.6708 ^#^
Moderate	52 (91)	5 (9)	57	
Well	6 (100)	0 (0)	6	
Histology type				
SQC	73 (94)	5 (6)	78	0.2987 ^&^
ADC	72 (90)	8 (10)	80	

^&^ Fisher’s exact test; ^#^ Chi Square test; *p* ≤ 0.05.

**Table 5 cells-13-01578-t005:** BARF1 expression and clinicopathological features of lung cancer.

Feature	BARF1 Expression		Total	*p*-Value
	(-)	(+)		
Total	30 (81)	7 (19)	37 (100)	
Smoking report				0.585 ^#^
Smoking	3 (75)	1 (25)	4	
Non-smoking	2 (100)	0 (0)	2	
Without report	25 (81)	6 (19)	31	
Differentiation				
Poor	22 (85)	4 (15)	26	0.464 ^#^
Moderate	7 (78)	2 (22)	9	
Well	0 (0)	0 (0)	0	
Without report	1 (50)	1 (50)	2	
Histology type				0.758 ^&^
SQC	15 (79)	4 (21)	19	
ADC	15 (83)	3 (17)	18	
EBV detection				
qPCR	30 (81%)	7 (19%)	37	0.06796 ^#^
IHC	26 (86%)	4 (14%)	30	
CISH	7 (54%)	6 (46%)	13	

^#^ Chi Square test for Table F by C; ^&^ Fisher’s exact test with mid-P method.

## Data Availability

The data presented in this study are available on request from the corresponding authors.

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
