# Peer review of "Epstein-Barr Virus BARF1 Is Expressed in Lung Cancer and Is Associated with Cancer Progression"

_cells, 2024, doi:10.3390/cells13181578_

Round 1
Reviewer 1 Report
Comments and Suggestions for Authors
The authors present the article entitled "EPSTEIN-BARR VIRUS BARF1 IS EXPRESSED IN LUNG CANCER AND IS ASSOCIATED WITH CANCER PROGRESSION" in which they evaluate the possible implication that EBV infection and lung cancer progression could have. In this sense, they evaluate the possible effects that the expression of the BARF1 gene could have on cellular processes associated with cancer progression, such as cell proliferation, migration and invasion.
In this sense, I consider that the authors consider their experiments as clear evidence of the participation of BARF1 in cancer development, which I consider a bit risky. To make this statement, I consider that more tests need to be carried out, including, for example, increasing their n.
According to the above, I have the following comments for the authors:
1. The authors try to establish a relationship between EBV infection and cancer development, however, their n is very small and other studies show a great variability in terms of its incidence. Given the above, how could you explain the implications of these numbers with the onset or development of lung cancer?
2. The authors use two cell lines, one normal and one derived from an adenocarcinoma. In this regard, what are the differences between them?, that is, in terms of chromosome number or expression of p53, c-myc, etc. Could these differences in genetic background explain, in part, the results of their assays on cancer progression?
3. Do the authors consider that the changes in proliferation, migration and invasiveness are specific to the exogenous expression of BARF1 and the genetic background of the cell lines? Or would the exogenous expression of BARF1 in any cell line give rise to the same results that the authors observed?
4. The authors propose a model in which the expression of BARF1 modifies some cellular processes (figure 9). However, it would have been interesting if they had evaluated the expression of the genes involved in these processes, such as p53, p21, Bax, Bcl2-xL. With the above, they would have better evidence of what they are proposing.
Minor observations
I think that the quality of the images should be improved, mainly the first 4. At least I don't see them with good resolution.
Moderate editing of English language required.
Author Response
REVIEWER: The authors try to establish a relationship between EBV infection and cancer development; however, their n is very small and other studies show a great variability in terms of its incidence. Given the above, how could you explain the implications of these numbers with the onset or development of lung cancer
Answer: Many thanks for this observation. The samples were 80 adenocarcinomas (AdCs) and 78 squamous cell carcinomas (SQCs), stored in the Pathological Anatomy Service of the National Thorax Institute, Santiago, Chile. This study is the first known survey and the biggest in South America. A sentence was added in the Discussion section. We found that EBV was present in 37 out of 158 (23%) lung carcinomas by PCR. In addition, nuclear EBNA1 (IHC) was found in 30/31 (97%) PCR positive lung carcinomas and moreover, EBERs (CISH) were found in 13/16 (81%) PCR positive lung carcinomas. This last method is the gold standard for EBV detection in clinical specimens. Thus, our analytical approach confirms the EBV presence in lung carcinomas from Chile. In addition, considering that BARF1 is a recognized epithelial oncogene, in this study we detected BARF1 transcripts in the lung tumors, and in vitro approaches demonstrated a correlation with increased tumorigenic properties (Migration, invasion and EMT). However, the role of EBV in this malignancy remains to be established and additional studies are warranted to characterize the specific role of EBV in lung cancer. For instance, a question is regarding the interaction between tobacco smoke and EBV, which we are addressing now in the laboratory. As tobacco smoke is a recognized carcinogen for lung cancer development, how tobacco components can modulate the EBV replication in lung cells, is an interesting concern for future research.
REVIEWER: The authors use two cell lines, one normal and one derived from an adenocarcinoma. In this regard, what are the differences between them? that is, in terms of chromosome number or expression of p53, c-myc, etc. Could these differences in genetic background explain, in part, the results of their assays on cancer progression
Answer: Many thanks for this observation. BEAS-2B, derived from normal bronchial epithelium, exhibits a diploid chromosomal complement, and typically expresses wild-type p53. In contrast, A549, originating from a lung adenocarcinoma, often displays chromosomal aberrations, including aneuploidy, and frequently harbors mutated or absent p53. Additionally, oncogenes like c-myc are often overexpressed in A549 cells. We added a sentence to declare these differences into the manuscript. Yes, this genetic background, at least in part, can explain the observed differences
REVIEWER: Do the authors consider that the changes in proliferation, migration and invasiveness are specific to the exogenous expression of BARF1 and the genetic background of the cell lines? Or would the exogenous expression of BARF1 in any cell line give rise to the same results that the authors observed?
Answer: Many thanks for this observation. The observed changes in proliferation, migration, and invasiveness in BEAS-2B and A549 cell lines upon BARF1 overexpression are likely influenced by a combination of the exogenous expression of BARF1 and the distinct genetic backgrounds of these cell lines. It is important to note that both BEAS-2B and A549 cell lines were extensively characterized prior to BARF1 transfection. This baseline characterization provided crucial information about the intrinsic properties of each cell line, including their growth rates, migratory potential, and invasive capacity. By comparing the phenotypes of the BARF1-overexpressing cells to their respective wild-type cell lines, we were able to confidently attribute any observed changes to the specific effects of BARF1. Previously in an article published by our group, we reported that BARF1 increases cell proliferation and migration of HPV16-positive cervical cancer cells. (Blanco R, et al. doi: 10.3390/microorganisms10050888). While our previous studies in cervical cancer have linked BARF1 expression to increased tumor aggressiveness, the current study extends these findings to lung cancer models. By utilizing well-characterized cell lines, we have been able to establish a causal relationship between BARF1 overexpression and the observed phenotypic changes.
REVIEWER: The authors propose a model in which the expression of BARF1 modifies some cellular processes (figure 9). However, it would have been interesting if they had evaluated the expression of the genes involved in these processes, such as p53, p21, Bax, Bcl2-xL. With the above, they would have better evidence of what they are proposing.
Answer. Many thanks for this comment. The ability of BARF1 is able to induce antiapoptotic proteins Bcl2-xL and Bax has been previously demonstrated in some cancer models (doi: 10.3892/ol.2018.8293; 10.1016/j.canlet.2005.06.023). Indeed, assessing the expression of these genes in our model would provide crucial mechanistic insights into how BARF1 modulates cellular processes like proliferation, migration, and apoptosis. For instance, analyzing the expression of p53 and its downstream targets could help us understand whether BARF1 influences the DNA damage response and cell cycle arrest. Similarly, examining the expression of Bax and Bcl-2xL would shed light on the potential impact of BARF1 on the intrinsic apoptotic pathway. While our current study focuses on the phenotypic consequences of BARF1 overexpression, incorporating a more in-depth molecular analysis, as suggested by the reviewer, would provide a more comprehensive understanding of the underlying mechanisms. We understand the importance of these additional analyses and we are planning to incorporate them into our future investigations.
REVIEWER: Minor observations
I think that the quality of the images should be improved, mainly the first 4. At least I don't see them with good resolution.
Answer: Many thanks for this observation: The quality of images were inproved.
REVIEWER: Comments on the Quality of English Language
Moderate editing of English language required.
Answer: Many thanks for this observation: We will improve the English language
Reviewer 2 Report
Comments and Suggestions for Authors
This seems to be the first South American study of EBV prevalence in lung cancer. The methods are well described and tables and figures are clear with good captions. The findings are similar to those in some other countries. The conclusion that EBV may be involved progression of a subset of lung carcinomas seems supported by this work.

Minor English corrections needed - see comments in attached file.
Author Response
Comments and Suggestions for Authors
This seems to be the first South American study of EBV prevalence in lung cancer. The methods are well described and tables and figures are clear with good captions. The findings are similar to those in some other countries. The conclusion that EBV may be involved progression of a subset of lung carcinomas seems supported by this work.
Comments on the Quality of English Language
Minor English corrections needed - see comments in attached file.
Answer: Many thanks for these observations. The English language concerns were improved according to the submitted instructions.
Reviewer 3 Report
Comments and Suggestions for Authors
Comments to authors
This manuscript describes the association of Epstein-Barr virus BARF1 and progression of lung cancer in Chili. For this purpose, they stably transfected a lung cancer cell line A549 and normal bronchial tissue BEAS-2B with an adenoviral plasmid expressing MSCV (control) or MSCVBARF1. They analyzed subsequent cell proliferation, migration, invasion, and epithelial-mesenchymal transition (EMT) in vitro. They found that BARF1, an early lytic protein frequently detected in EBV-pos carcinomas, increased migration, proliferation, invasion and EMT in the normal bronchial cells. While EBV is detected in 95% of the human population, BARF1 was detected in 46% of the EBV-pos lung cancers by RT-qPCR. They concluded that EBV is frequently found in lung cancer from Chili with expression of BARF1, suggesting that this viral protein may be involved in EBV-associated lung cancer progression.
Overall, this is an interesting study. However, the association of BARF1 and cancer progression is moderately innovative. Testing the BARF1-transfected normal brionchial cells in nude or immune competent mice would significantly increase the novelty of the study. While BARF1 alone may not induce cancer, it may provide information which genes are turned on or off and whether secretion of cytokines like IL-10, IL-6, TGFbeta has been initiated by BARF1 in vivo. It is interesting to see that proliferation, invasion and EMT increased, but in vitro and in vivo can be very different. I agree that it is suggestive that these viral proteins may be involved in cancer progression, but this has been suggested a long time ago. Also, transfecting a BARF1-negative lung cancer cell line with BARF1 gene and then analyzing what happens with the cancer in vivo could be very interesting.
In summary, lack of causal in vivo data reduces the strength of the study.
Minor
· Please explain whether the A549 cell line expresses EBV or BARF1 naturally.
· Please explain what the 2 BNRF1 bands are in Fig 1. Are they both BNRF1?
Author Response
REVIEWER: This manuscript describes the association of Epstein-Barr virus BARF1 and progression of lung cancer in Chili. For this purpose, they stably transfected a lung cancer cell line A549 and normal bronchial tissue BEAS-2B with an adenoviral plasmid expressing MSCV (control) or MSCVBARF1. They analyzed subsequent cell proliferation, migration, invasion, and epithelial-mesenchymal transition (EMT) in vitro. They found that BARF1, an early lytic protein frequently detected in EBV-pos carcinomas, increased migration, proliferation, invasion and EMT in the normal bronchial cells. While EBV is detected in 95% of the human population, BARF1 was detected in 46% of the EBV-pos lung cancers by RT-qPCR. They concluded that EBV is frequently found in lung cancer from Chili with expression of BARF1, suggesting that this viral protein may be involved in EBV-associated lung cancer progression. Overall, this is an interesting study. However, the association of BARF1 and cancer progression is moderately innovative. Testing the BARF1-transfected normal brionchial cells in nude or immune competent mice would significantly increase the novelty of the study. While BARF1 alone may not induce cancer, it may provide information which genes are turned on or off and whether secretion of cytokines like IL-10, IL-6, TGFbeta has been initiated by BARF1 in vivo. It is interesting to see that proliferation, invasion and EMT increased, but in vitro and in vivo can be very different. I agree that it is suggestive that these viral proteins may be involved in cancer progression, but this has been suggested a long time ago. Also, transfecting a BARF1-negative lung cancer cell line with BARF1 gene and then analyzing what happens with the cancer in vivo could be very interesting.
In summary, lack of causal in vivo data reduces the strength of the study.
Answer: Many thanks for this observations. The proposed in vivo experiments, including testing BARF1-transfected normal bronchial cells in nude or immunocompetent mice, would undoubtedly enhance the translational impact of our findings. Additionally, investigating the in vivo effects of BARF1 on gene expression and cytokine secretion would provide crucial insights into its potential role in tumorigenesis and immune modulation.
While we recognize the importance of these experiments, the scope of the current study primarily focused on establishing the in vitro effects of BARF1 on cellular behavior. We believe that conducting these in vivo studies would require a significant expansion of our experimental design and resources. Therefore, we plan to incorporate these valuable suggestions into future research endeavors to further elucidate the role of BARF1 in lung cancer development and progression.
Minor
REVIEWER: Please explain whether the A549 cell line expresses EBV or BARF1 naturally.
Answer: Many thanks for this observation. A549 cells (ATCC. CCL-185) originate from human lung adenocarcinoma cells and they are free of infectious agents, they do not contain genetic material from viruses, and consequently, they do not express BARF1.
REVIEWER: Please explain what the 2 BNRF1 bands are in Fig 1. Are they both BNRF1?
Answer: Many thanks for this observation. In Figure 1, the agarose gel was employed to visualize a PCR product of approximately 95 base pairs. Two distinct bands are apparent: an upper band corresponding to the expected PCR product, and a lower band. The lower band is a common artifact in PCR known as a primer-dimer. Primer dimers are short DNA fragments formed when PCR primers anneal to each other due to complementary sequences at their 3' ends. These dimers can be amplified during the PCR reaction, resulting in a visible band on the gel. It's important to note that primer dimers are non-specific products and do not represent the target sequence of interest. Therefore, for the purposes of this analysis, only the upper band, corresponding to the expected 95 bp PCR product, is relevant. The presence of primer-dimers, while a common occurrence in PCR, does not affect the interpretation of the results regarding the target gene of interest.
Round 2
Reviewer 1 Report
Comments and Suggestions for Authors
The authors responded to my comments.
Author Response
Many thanks for the observations to improve this manuscript.
Reviewer 3 Report
Comments and Suggestions for Authors
Comments to authors-2
I agree with analyzing causal effects in a follow up paper, but the value of this manuscript is less strong. The role of viral sequences in cancer progression is interesting. The manuscript has been improved.
There is one minor issue. Please, instead just showing the bar graph in each figure include also the individual points.
I have no further comments.

Author Response
Many thanks for this observation. The graphs were modified including the individual points (migration and invasion).